# Epigenetic Regulation of Immunotherapy Response in Triple-Negative Breast Cancer

**DOI:** 10.3390/cancers13164139

**Published:** 2021-08-17

**Authors:** Pere Llinàs-Arias, Sandra Íñiguez-Muñoz, Kelly McCann, Leonie Voorwerk, Javier I. J. Orozco, Miquel Ensenyat-Mendez, Borja Sesé, Maggie L. DiNome, Diego M. Marzese

**Affiliations:** 1Cancer Epigenetics Laboratory at the Cancer Cell Biology Group, Institut d’Investigació Sanitària Illes Balears (IdISBa), 07120 Palma, Spain; pere.llinas@ssib.es (P.L.-A.); sandra.iniguez@ssib.es (S.Í.-M.); miquelarash.ensenat@ssib.es (M.E.-M.); borja.sese@ssib.es (B.S.); 2Division of Hematology/Oncology, Department of Medicine, David Geffen School of Medicine, University of California Los Angeles, Los Angeles, CA 90095, USA; KMccann@mednet.ucla.edu; 3Division of Tumor Biology & Immunology, The Netherlands Cancer Institute, 1066CX Amsterdam, The Netherlands; l.voorwerk@nki.nl; 4Saint John’s Cancer Institute, Providence Saint John’s Health Center, Santa Monica, CA 90404, USA; javier.orozco@providence.org; 5Department of Surgery, David Geffen School of Medicine, University California Los Angeles (UCLA), Los Angeles, CA 90024, USA; mdinome@mednet.ucla.edu

**Keywords:** epigenetics, TNBC, immunotherapy, cancer, breast cancer, epigenetic drugs, immune system, immune checkpoint

## Abstract

**Simple Summary:**

Triple-negative breast cancer (TNBC) outcomes are improving since the implementation of immunotherapy. However, objective response rates are still limited to a select group of patients. This is partly due to TNBC intrinsic immune evasive mechanisms and the lack of proper tumor microenvironment immune system activation. Dynamic epigenetic modifications contribute to immune surveillance and immune escape in cancer and can be reverted through epigenetic drugs. This review summarizes the epigenetic changes in TNBC cells and their contribution to the cancer cell–immunity cycle. Furthermore, it also describes how epigenetic drugs may provide novel biomarkers for immunotherapy and enhance the immune response. This manuscript lists the current clinical trials using epigenetic drugs alone or combined with either immune checkpoint inhibitors or small molecules.

**Abstract:**

Triple-negative breast cancer (TNBC) is defined by the absence of estrogen receptor and progesterone receptor and human epidermal growth factor receptor 2 (HER2) overexpression. This malignancy, representing 15–20% of breast cancers, is a clinical challenge due to the lack of targeted treatments, higher intrinsic aggressiveness, and worse outcomes than other breast cancer subtypes. Immune checkpoint inhibitors have shown promising efficacy for early-stage and advanced TNBC, but this seems limited to a subgroup of patients. Understanding the underlying mechanisms that determine immunotherapy efficiency is essential to identifying which TNBC patients will respond to immunotherapy-based treatments and help to develop new therapeutic strategies. Emerging evidence supports that epigenetic alterations, including aberrant chromatin architecture conformation and the modulation of gene regulatory elements, are critical mechanisms for immune escape. These alterations are particularly interesting since they can be reverted through the inhibition of epigenetic regulators. For that reason, several recent studies suggest that the combination of epigenetic drugs and immunotherapeutic agents can boost anticancer immune responses. In this review, we focused on the contribution of epigenetics to the crosstalk between immune and cancer cells, its relevance on immunotherapy response in TNBC, and the potential benefits of combined treatments.

## 1. Introduction

Triple-negative breast cancer (TNBC), which represents 15–20% of breast cancers (BC), is classified based on the exclusion criteria of lack of estrogen and progesterone receptor expression and absence of human epidermal growth factor receptor 2 (HER2) overexpression. TNBC is particularly aggressive, with a higher probability of metastatic progression and a lack of effective targeted therapies [1]. Given its classification method of BC, TNBC is a heterogeneous malignancy that encloses tumors with different histopathological and molecular features [2]. Thus, several studies focused on the classification of TNBC subtypes. The first classification approach took advantage of the microarray technology to identify six different TNBC subtypes: basal-like 1 (BL1), basal-like 2 (BL2), mesenchymal (M), mesenchymal stem-like (MSL), immunomodulatory (IM), and luminal androgen receptor (LAR) [3]. TNBC subclassification allows for a better understanding of this disease, yet, it has shown limitations in that it fails to predict specific and effective treatments for the TNBC subtype.

This has encouraged the search for new therapeutic alternatives, such as PARP inhibitors and, more recently, immune checkpoint inhibitors (ICI). Thus, different regimens combining immunotherapy and chemotherapy are currently approved by the Food and Drug Administration (FDA) and the European Medicines Agency (EMA) for TNBC patients: The combination of nab-paclitaxel and atezolizumab [4] and pembrolizumab in combination with chemotherapy has demonstrated positive results in metastatic TNBC [5]. It was recently shown that neoadjuvant immunotherapy combined with chemotherapy shows increased pathological complete response rates in early-stage TNBC [6,7]. Nevertheless, the iRECIST objective response rate (ORR) for unselected cohorts of TNBC patients remains below 10% [8]. The adaptive phase II TONIC clinical trial revealed that the induction of the tumor and immune system cells with low doses of doxorubicin significantly improves ICI response in advanced TNBC patients (ORR: 35%), independently of the tumor mutational burden [9]. This study highlighted the importance of the dynamic phenotypic adaptation of TNBC and immune cells for immune response.

Unlike other solid tumors, TNBC displays a low frequency of genetic alterations, highlighting the relevance of epigenetic modifications during cancer progression and establishing aggressive phenotypes. Two well-established epigenetic modifications involve DNA methylation and histone modifications. DNA methylation mainly occurs on the 5th position of cytosines, followed by guanosines (also referred to as 5mC). The catalog of histone modifications is far more complex since it involves more than a single chemical group and position. These fine-tuned chemical modifications are controlled by a set of enzymes with rising importance in cancer research: writers are involved in depositing chemical marks, erasers are responsible for the removal, and readers recognize the epigenetic code and recruit other proteins [10]. In general, epigenetic alterations involving DNA or histone modifications are summarized based on the impact on the associated gene(s). Thus, gene promoter hypermethylation or repressive histone marks are usually associated with silencing of the nearby genes (i.e., tumor suppressor genes [11] such as the *BRCA1* gene [12]).

Nevertheless, beyond gene promoters, epigenetic alterations can affect gene regulatory elements (GRE) such as enhancer and insulator elements determining activation of cancer-associated gene expression programs and global DNA hypomethylation leading to genomic instability and the potential oncogene reactivation [13]. An overview of epigenetic mechanisms is summarized in Figure 1. However, epigenetic modifications are not only being involved in tumor suppressor gene silencing and oncogene reactivation. Due to the dynamic nature of epigenetic modifications, it is important to consider the epigenetic landscape of the tumor microenvironment (TME), its adaptation to tumor-induced changes, and its contribution to immunotherapy [14]. This will allow for understanding the interplay between TNBC and immune cells and identifying biomarkers to better stratify patients for immunotherapy or complementary therapeutic targets. In addition, chemical removal of aberrant epigenetic marks using small molecule inhibitors may enhance the immune response by expressing immunogenic antigens and reactivating transposable elements [15]. In this context, understanding the epigenetic mechanisms involved in TNBC immune-suppressive pathways and immune cells activation represents an opportunity to improve the selection of TNBC patients for immunotherapy and explore alternative and complementary therapeutic targets. In this review, we cover the importance of epigenetic mechanisms in immunotherapy response in patients with TNBC.

## 2. Epigenetic Relevance on Antitumor Immune Response

Both innate and adaptive immunity mediates the cancer immune response. Innate immunity is performed by natural killers (NKs), dendritic cells (DC), eosinophils, and tumor-associated macrophages (TAMs) [16]. The activation of these cells involves a fine-tuned epigenetic modulation of the gene expression program [17,18,19]. Conversely, epigenetic alterations contribute to immunosuppression by innate immune cells. For instance, epigenetic silencing through promoter hypermethylation of NKG2D ligands impairs NK function in acute myeloid leukemia [20]. Focusing on TNBC, TAM polarization into M2 protumorigenic macrophages is mediated by miR-200C [21], and TAM infiltration is associated with a higher risk of distant metastasis in TNBC [22].

In 2013, Chen and Mellman proposed the cancer–immunity cycle (Figure 2), consisting of sequential steps describing adaptive immunity against tumor cells. Briefly, cancer cells express aberrant antigens (step 1) called tumor-associated antigens (TAAs), which are released into the TME after cell death. TAAs are captured by antigen-presenting cells (APCs), such as DCs, that migrate to the lymph nodes, where they present the TAAs through the major histocompatibility complex (MHC) to naïve T-cells (step 2). This presentation drives T-cell priming and activation (step 3). Finally, activated cytotoxic T lymphocytes (CTLs) leave the lymph nodes and enter the bloodstream (step 4), where chemokine detection leads to extravasation into the TME (step 5). There, cancer cells must be recognized by CTLs (step 6) and killed (step 7) [23]. Although epigenetic changes are involved in all the listed steps of the cancer-immunity cycle, our review emphasizes the epigenetic regulation of those steps involving tumor cells (steps 1, 5, 6, 7; Figure 2). Epigenetic changes involving DC maturation and CTL activation have been reviewed elsewhere [24,25].

### 2.1. Aberrant Antigen Expression

The tumor-specific global hypomethylation and chromatin organization promote the reactivation of epigenetically silenced genes in healthy tissues [26]. Some of these genes may trigger an immune response since they act as neoantigens [27] or immunogenic cancer-testis antigens (CTAs), which act as TAAs. CTAs are downregulated in somatic adult tissues but aberrantly expressed in different malignancies. In addition, treatment with demethylating agents promotes CTA re-expression, suggesting that DNA methylation is responsible for silencing in somatic tissues [28]. Similarly, gene promoter hypermethylation has been proposed as a mechanism able to silence neo-antigen expression [27]. In TNBC, specifically, different studies have identified the upregulation of *NY-ESO-1*, *MAGE-1*, *WT1*, and *SPANXB1* [29,30,31]. Thus, aberrant CTA expression in TNBC has become a therapeutic opportunity. Clinical trials based on vaccines targeting CTAs are currently being performed in solid tumors [32]. Beyond their potential relevance as therapeutic targets, CTAs also have an impact on TNBC tumor biology. For example, *SPANXB1* expression was associated with increased migration and invasion abilities. In addition, its mRNA and protein levels were negatively correlated with the metastasis suppressor gene *SH3GL2* [31].

Global hypomethylation also disturbs the immune response through the abnormal expression of endogenous retroviruses (ERV) [33]. Although ERV activation drives the expression of oncogenes [34], this alteration also produces double-stranded RNA molecules, which promote an IFN-mediated viral mimicry response [35], activating the innate immune response. Nevertheless, cancer cells may overcome this setback. For example, taxane-resistant TNBC cell lines display global DNA hypomethylation, but an epigenetic switch prevents ERV activation. EZH2, a histone writer whose upregulation is mainly observed in TNBC [36], represses ERV sequences through H3K27me3 histone mark deposition, avoiding viral mimicry and eluding the immune system [37]. Viral mimicry and its further antiviral signaling have been observed after spliceosome-targeted therapies in TNBC, promoting innate and adaptive immune responses and providing new therapeutic strategies [38]. Taken together, the modulation of the TNBC epigenome may promote viral mimicry responses or tumor-associated antigen expression, which in turn may trigger an immune response.

### 2.2. Chemokine-Mediated Recruitment

After antigen presentation and T-cell activation in the lymph nodes, CD8+ cytotoxic lymphocytes (CTLs) migrate through blood vessels. There, CTLs may detect a chemokine gradient and extravasate into the TME. The number of CTLs recruited at the tumor site is a predictor of immunotherapy response across different cancers [39], mediated by chemokine production on the tumor site [40]. Furthermore, high levels of the C-X-C motif chemokine ligands 9 and 10 (CXCL9 and CXCL10), C-C motif chemokine ligand 5 (CCL5), and IFN-y are associated with enhanced levels of CTLs in the TME [41]. This recruitment correlates with increased survival and lower levels of cancer metastasis in cancer patients [42]. Using TNBC cell lines, Qin et al. elucidated the possible mechanisms that underlie epigenetic dysregulation in the activity of chemokines and how they blocked antitumor immune cells’ circulation. They concluded that an epigenetic modifier, Lysine-Specific Demethylase 1 (LSD1), altered the cell landscape in TNBC through chemokine silencing, especially *CCL5* and *CXCL10* expression [43].

### 2.3. T-Cell Recognition & Antigen Presentation

Inhibition of the immune system is closely interconnected with tumor progression and development. The downregulation of antigen processing and presentation, especially the lack of MHC class I expression, allows tumor cells to evade immune surveillance [44]. Downregulation of MHC class I occurs more commonly than full elimination since full depletion makes cancer cells sensitive to the effect of NK cells via non-classical MHC molecules [45]. Moreover, epigenetic dysregulation of antigen processing and the presentation of machinery-related genes has been observed in cancer cells. These alterations include the depletion of the MHC class I transactivator *NLRC5* and the HLA class II-chaperone *CD74* through DNA methylation in different types of cancer [46,47].

MHC class I impairment has been observed as a mechanism of immunotherapeutic resistance, particularly in the TNBC apocrine subtype [48] and metastatic TNBC [49]. In metastatic TNBC, high expression of the transmembrane protein MAL2 diminishes the level and stability of the antigen-loaded MHC class I on the cell membrane, promoting an ineffective antigen presentation and consequently limited recognition by CD8+ T-cells [50].

In TNBC, aberrant overexpression of the MHC class II pathway is associated with increased T-cell infiltration and prolonged progression-free survival (PFS) [51]. In addition, the expression of MHC class II molecules, such as HLA-DR, in tumor tissue has been linked to the presence of tumor-infiltrating lymphocytes (TILs) and high expression of *CD4*, *CD3D*, and *CD8A* [52]. MHC class I and MHC class II expression are regulated through promoter DNA methylation of their coding HLA genes. An inverse correlation exists between the mRNA expression levels and DNA methylation on the surrounding region of transcription start sites of HLA genes in BC. Moreover, hypermethylation of HLA promoters also correlates with decreased *CD8A* mRNA levels [53].

Epigenetic modulation using the DNA Methyltransferase inhibitor (DNMTi) guadecitabine, a next-generation hypomethylating agent, promotes effective CD8 T-cell responses via increased MHC class I expression, increased IFN-y secretion, and recruitment of CTLs into the TME. Thus, DNMTi treatment might have countless effects on the interplay between tumor and immune systems, such as immune response activation and demethylation of MHC class I [53,54]. Regarding MHC class I, studies have identified the melanoma-associated antigen-A11 (MAGE-A11) peptides presented by HLA class I molecules [55,56,57]. MAGE-A11 is a CTA usually expressed in BC and related to poor prognosis. Therefore, its induction is relevant for the recognition and killing of BC cells by CTLs. Furthermore, MAGE-A11 antigens induced cytotoxicity on MAGE-A11-positive TNBC cells by effector CTLs [58].

Different alterations may lead to the release of double-stranded DNA into the cytosol. These events activate the cGAS-STING pathway, which provokes a signaling cascade that enhances the transcriptional expression of type I interferons and other immune-stimulatory genes, promoting an immune response [59]. Given its role as an immune-stimulatory pathway, cancer cells tend to downregulate STING expression levels. For instance, KDM5-mediated histone demethylation and transcription repression was observed in BC cell lines. However, this epigenetic signaling has not been studied in TNBC cell lines or in vivo models. [60]. Given this relevance, the development of human STING agonists has arisen. In fact, the administration of STING agonists resensitized TNBC immunocompetent mice against ICI [61,62].

### 2.4. Cancer Cell Elimination

Once recognized, CTLs can eliminate cancer cells, whereas immune checkpoints (ICs) can compromise this immune response. ICs include different inhibitory pathways that control the intensity and duration of immune responses [63]. It is well known that in the TME, cancer cells can escape the cytotoxic effect of T-cells by activating IC pathways [64]. The best-described strategies to de-activate T-cells are the binding of PD-1 on T-cells to programmed death-ligand 1 (PD-L1) on cancer cells and APCs [65] and CTLA-4 on T-cells to CD80/86 on APCs [66]. Different IC proteins, such as TIM-3, CTLA-4, and LAG-3, are upregulated in primary BC through epigenetic mechanisms, including DNA hypomethylation and decreased repressive histone marks H3K27me3 and H3K9me3, in the promoter regions [67]. Regarding PD-L1 regulation, it is well established that DNA methylation affects its expression in different cancers, including melanoma and gastric cancer [68,69]. However, PD-L1 seems to be completely demethylated in BC, and its regulation is based on histone modifications [67,70]. The upregulation of different immune checkpoint proteins is supported by aberrant epigenetics events that may be corrected using epigenetic drugs. Thus, histone deacetylase inhibitors (HDACi) and DNMTi can revert the aberrant expression and restore antitumor immunity [71].

In this context, Terranova-Barberio et al. evaluated the effect of HDACi combined with ICI to enhance immunotherapy responses in TNBC in vivo. They observed that HDACi upregulates PD-L1 and HLA-DR expression in TNBC cells and improves the response to PD-1/CTLA-4 blockade in a TNBC mouse model, decreasing tumor growth and improving survival. This was associated with increased T-cell tumor infiltration and downregulation of CD4+ and FOXP3+ T-cells [72].

Beyond CTLA-4 and PD-1/PD-L1, there is an increasing interest in inhibitory receptors as potential targets in immunotherapy. CD155 and CD112 are expressed on the tumor cell surface. These markers interact with the T-cell immunoreceptor with Ig, and ITIM domains (TIGIT) found on NK, CD8+, and CD4+ T-cell membranes [73]. TIGIT is poorly expressed on naïve T-cells, but it is overexpressed after promoter hypomethylation and FOXP3 binding [74]. The overexpression of TIGIT ligand CD155, encoded by the PVR gene, has been reported in metastatic BC compared with normal tissue [75].

Furthermore, TNBC displays higher CD155 expression than other BC subtypes [75,76]. Interestingly, an active enhancer has been found close to the *PVR* promoter, which may explain this phenotype (Figure 3). This overexpression was observed in all BC subtypes, being more prevalent in TNBC. CD155 expression correlated with a worse prognosis in BC, pointing out its relevance on new treatments and the outcome prediction [75]. B7-H3 is another promising target in TNBC. Its enhanced expression on TAMs and cancer cells in TNBC patients promoted a pro-angiogenic state and correlated with a worse prognosis [77]. The antibody-mediated inhibition of this immune receptor enhanced the therapeutic effect of PD-1 blockade in TNBC murine models [78]. An in vitro study performed on a TNBC cell line revealed that B7-H3 knockdown reduced glycolytic activity and sensitized cells against AKT/mTOR inhibitors [79]. Different studies point out different epigenetic mechanisms disrupted cancer progression, promoting increased B7-H3 expression levels [80,81,82].

## 3. Other Connections between Epigenetics and Immunotherapy in TNBC

### 3.1. Tumor Microenvironment

The TME composition also impacts immunotherapy response. Tumors can be classified as ‘hot’ when they exert T-cell infiltration and inflammation or ‘cold’ when enriched in immunosuppressive cells, such as TAMs, myeloid-derived suppressor cells (MDSCs), and regulatory T-cells (Tregs). Hot tumors show a better response against IC therapies. For that reason, several strategies—including epigenetic reprogramming—aim to turn ‘cold’ tumors into ‘hot’ [71] (Figure 4).

Consequently, the presence of CD8+ infiltrating T-cells is associated with a good prognosis and a better response against anti-PD-L1 treatments in TNBC and other cancer subtypes [83,84,85]. *EZH2* overexpression correlates with decreased infiltration of CD8+ T-cells, translating into a ‘cold’ phenotype. Persistent antigen stimulation of CD8+ T-cells and an immunosuppressive TME contribute to CD8+ T-cell ‘exhaustion’, which is translated into the activation of Pdcd1 (encoding PD-1) and Il-10 signaling pathways [86]. This alteration has been observed in melanoma murine cell lines [87]. As far as we know, no study has covered the epigenetic role in this state of cell dysfunction, focusing on TNBC.

MDSCs also generate an immunosuppressive environment through many mechanisms, including nutrient depletion and Tregs recruitment [88]. In addition, this population contributes to the development of premetastatic niches [89]. Interestingly, low doses of epigenetic treatments impaired this population in TNBC murine models [90] and disrupted the metastatic niche formation in BC [89]. Eosinophils display pleiotropic effects on tumor sites since they can produce and secrete cytotoxic proteins and angiogenic and matrix-remodeling factors [91]. In TNBC, relative eosinophil count has been associated with a lower relapse rate, suggesting tumor-inhibiting phenotype in this particular malignancy [92]. Cancer cells can reprogram the TAMs expression patterns through different mechanisms. TNBC cell lines displayed paracrine signaling that epigenetically activated the *ID4* promoter region in cancer cells but also in TAMs. This crosstalk promoted the activation of an angiogenic program, partially sustained by the downregulation of miR-15b/107 in TAMs [93,94]. The connections between TAMs and anti-PD-1/PD-L1 therapies are comprehensively reviewed by Santoni et al. [95].

### 3.2. Metabolic Rewiring

Metabolic reprogramming is another hallmark of cancer. Despite its intrinsic heterogeneity, TNBC displays a higher glucose dependence when compared with other BC subtypes, a feature partially sustained by epigenetic alterations. These alterations include promoter hypermethylation of the gluconeogenic enzyme FBP1 [96] and HIF-1α stabilization by the lncRNA LINK-A, among others [97]. This glucose dependence is translated into an increase in lactate production even in normoxic conditions, a phenomenon known as the “Warburg effect” [98]. The acidification of the TME and the lactate release inhibit the immune response [99]. Aerobic glycolysis also promotes MDSC recruitment, which contributes to immune suppression [100]. In addition, lactate was recently identified as a novel histone posttranslational modification [101]. Further research may highlight the relevance of epigenetic mechanisms in the pH regulation in TME and its potential role in immune response [102].

The kynurenine pathway (KP) also couples epigenetic alterations and immune response. During inflammation, IFN-gamma induces indoleamine 2,3-dioxygenase (IDO) expression in cancer cells and MDSCs, the first enzyme of the KP [103]. *IDO* overexpression occurs preferentially on TNBC with basal-like subtypes [104,105]. It seems to be negatively regulated by promoter hypermethylation on ER+ BC [106]. IDO enzymatic activity may promote tryptophan depletion and immunosuppressive metabolites synthesis, inhibiting T-cell activity and inducing immune tolerance [107,108].

### 3.3. Epithelial-to-Mesenchymal Transition

Many epigenetic alterations are associated with epithelial-to-mesenchymal transition (EMT) in TNBC, which have been deeply discussed by Khaled et al. [109]. The authors summarize different epigenetic mechanisms—including DNA methylation, histone modification, and long non-coding RNA (lncRNA) interactions that promote EMT and metastasis. Interestingly, EMT has also been identified as a resistance mechanism to immunotherapy in TNBC [110]. Among other mechanisms, the axis miR-200/*ZEB1*, which controls the metastasis program, promotes PD-L1 expression [111].

## 4. Better Together, the Combination of Epigenetic Inhibitors and Immunotherapy

Since their approval, epigenetic drugs have demonstrated efficacy in cancer treatment, especially in hematological malignancies [112]. Clinical adoption of epigenetic drugs may increase during the following years, fueled by their potential role in preventing metastasis [89] and potential combination with immunotherapy [113]. Epigenetic modulation of molecular pathways involved in the cancer cell immunity cycle (Figure 2) through inhibitors against writers, readers, and erasers (Figure 1) promote a decrease in immune evasion and sensitize cancer cells against ICIs [114]. DNMTi and HDACi promote upregulation of CTAs, PDL-1/PD-L2 and MHC-I-related genes expression, the reactivation of repetitive elements, and an increase in the chemokine expression and release [15]. 

In addition to the effect on cancer cells, low doses of epigenetic drugs impact different immune cell populations: HDAC6 inhibition promotes the activation of naïve T-cells [115], whereas Class I HDACi increase the response of T CD8+ and NK cells [116]. Treg lymphocytes activity can also be modulated through epigenetic drugs. Tregs display an immune-suppressive program, which is orchestrated by the transcription factor FOXP3. Thus, the inhibitor-mediated decrease of FOXP3 instability through the inhibition of the histone acetyltransferase EP300 may contribute to reestablishing the immune response [117,118].

DNMT1, whose upregulation on all BC subtypes correlates with a worse prognosis [119], could be modulated with decitabine, a DNMTi [120]. Decitabine efficacy is being tested with carboplatin and in combination with pembrolizumab, an anti-PD-1 antibody (NCT03295552 and NCT02957968, respectively). Furthermore, the reversibility of estrogen receptor depletion is being tested by combining decitabine and tamoxifen (NCT01194908). 5’-azacytidine, one of the first DNMTi, is being tested alone or combined with entinostat, an HDAC I inhibitor (NCT01292083 and NCT01349959, respectively). Clinical trials involving epigenetic drugs alone or combined with either immunotherapy or chemotherapy are listed in Table 1.

Entinostat has been tested in combination with first-line treatments such as doxorubicin [121] and immunotherapy. Entinostat enhanced the effect of an IL-15 agonist and a vaccine against TAAs in TNBC murine models [122] and displayed synergistic effects when combined with a PD-L1 inhibitor through its effect on MDSCs [123]. Clinical trials involving this epigenetic drug alone (NCT03361800) or in combination with atezolizumab (a PD-L1 inhibitor; NCT02708680) are being conducted. In addition, another HDAC I inhibitor called vorinostat is being clinically tested (NCT01695057). Beyond the first generation of epigenetic drugs targeting DNMTs and HDACs, more recently, inhibitors for bromodomains have been developed. *BRD4* was identified as a potential target in TNBC [124]. Since then, preclinical and clinical studies have highlighted that chemical inhibition of this BET bromodomain is an efficient treatment in TNBC patients [125,126,127]. Moreover, novel approaches using BET-proteolysis targeting chimeric compounds (BET-PROTACs), which allosterically inhibit BET bromodomains and bind a ubiquitin ligase, have been tested in TNBC with promising results, even in BET-resistant tumors [128]. Infiltration of TAMs in the TME correlates to BET inhibitor resistance in TNBC [129]. Furthermore, a recent study showed that BRD4 blockade impairs PD-L1 expression in TNBC [130].

## 5. Future Perspectives

The presence of epigenetic alterations modulates the immune response in TNBC, impairing antigen expression, decreasing chemokine production, developing an immunosuppressive TME, and promoting tumor survival through promoting immune-evasive mechanisms. Epigenome signatures involved in immune escape may represent an opportunity to identify complementary therapies and patients that may not respond to ICIs, as tested in other cancer types, such as non-small cell lung cancer [131]. In addition, these new approaches may complement the current biomarkers involved in response to immunotherapy by building integrative nomograms including epigenetic features and additional biomarkers, such as tumor neoantigens, intratumor heterogeneity, tumor metabolism, tumor-immune microenvironment gut microbiome, and checkpoints targets. These include PD-L1 protein expression and a fraction of high PD-1 mRNA [84]. Altogether, the interaction of these parameters may be depicted in individual patients as ‘cancer immunograms’ that will allow a more personalized stratification to guide immunotherapy decision-making [132].

Furthermore, the reported benefits of epigenetic drugs combined with immunotherapy might lead to strategies to further improve immunotherapy in TNBC patients. We also believe that the research focusing on cancer epigenetics will move to key GRE beyond promoters recently reported in TNBC [133]. In fact, key IC-related genes display enhancer elements close to the promoter region with potential involvement in immune response, as described in Figure 3. The characterization of these elements will provide a better understanding of cancer biology behind immune response and may provide new therapeutic targets. Taken together, growing evidence supports that epigenetic modulation is relevant for immunotherapy, and its clinical implementation may be a turning point to improve the outcome of patients with the most aggressive type of BC.

## Figures and Tables

**Figure 1 cancers-13-04139-f001:**
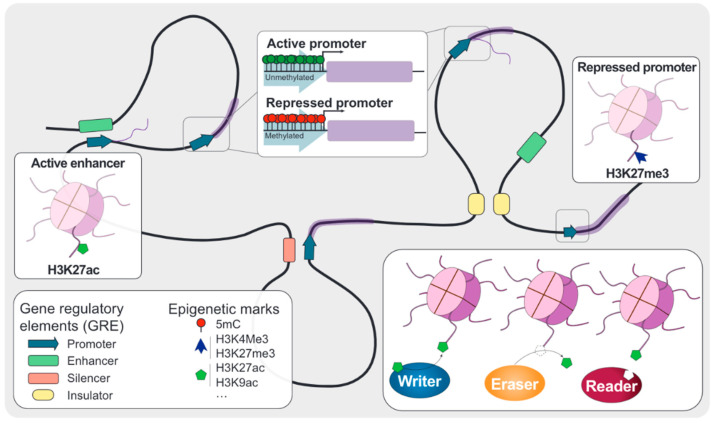
Overview of epigenetics. Gene regulatory elements (GRE) are defined according to their location and effect on the associated gene expression. Promoters are located close to the transcription start site of their associated genes and facilitate the transcription machinery deposition. This deposition can be aided or blocked by distant GRE called enhancers and silencers, respectively. Chromatin architecture is dynamically regulated by another group of GRE called insulators, which mediate in the topological associating domains (TADs) formation and further contributing to gene expression. DNA and histone modifications are tightly regulated by three different groups of proteins: writers, which are involved in the deposition of these chemical marks; erasers, which are responsible for removing these modifications; and readers, which can recognize the epigenetic code and recruit other proteins.

**Figure 2 cancers-13-04139-f002:**
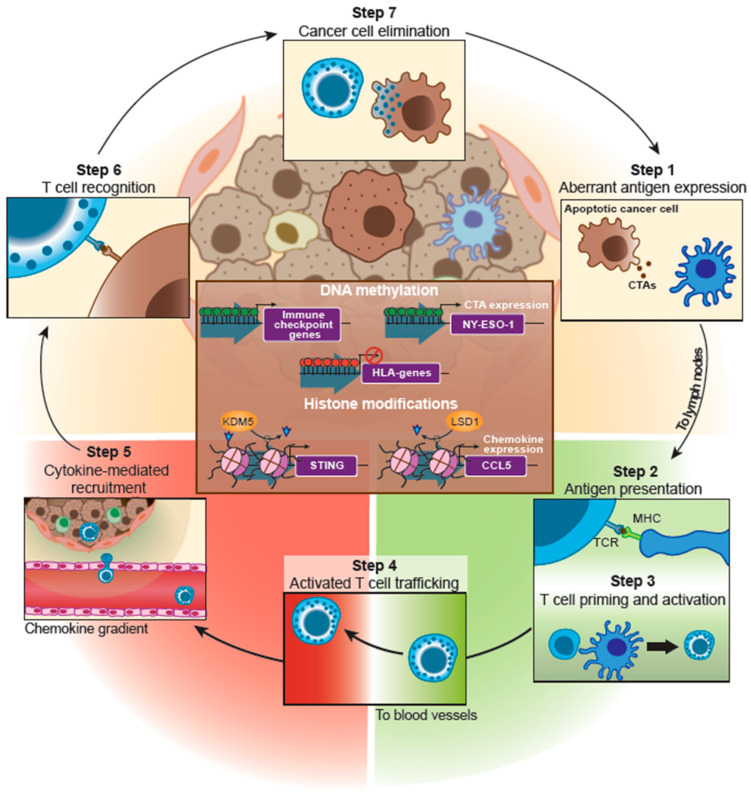
Epigenetic regulation of cancer–immunity cycle in TNBC. This scheme shows epigenetic mechanisms involved in TNBC immune escape (brown square in the middle). It also includes the different steps of the cancer immunity cycle. Scheme based on Chen and Mellman (2013) cancer-cell immunity cycle [23]. Abbreviations: CTAs (Cancer testis antigens), TCR (T-cell receptor), MHC (Major Histocompatibility complex).

**Figure 3 cancers-13-04139-f003:**
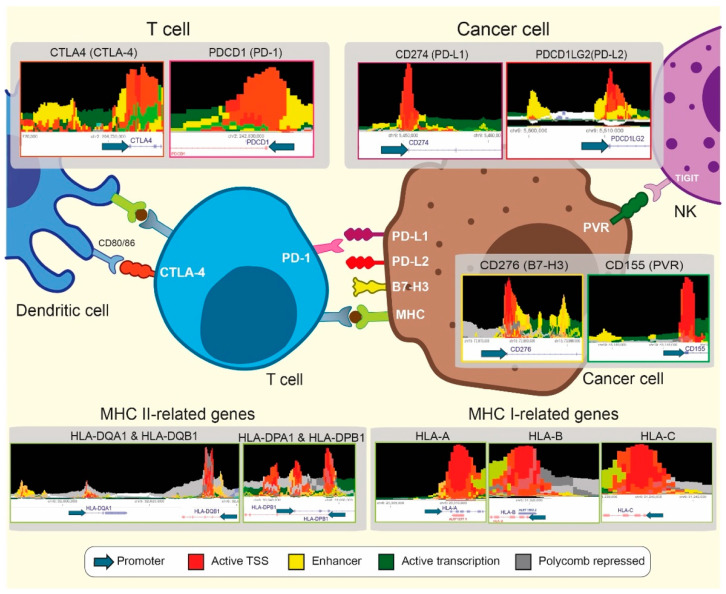
Therapeutic opportunities combining epigenetics and immunotherapy. Different GRE, including promoters and enhancers, have been identified close to the transcription start site (TSS) of IC-related genes using the Epilogos visualization tool (https://epilogos.altius.org/, accessed on 29 April 2021). A total of 127 human samples from the Roadmap Consortium were considered when cancer cell-expressing genes were interrogated. In contrast, the blood & T cells category was selected from T cell and NK expressing genes.

**Figure 4 cancers-13-04139-f004:**
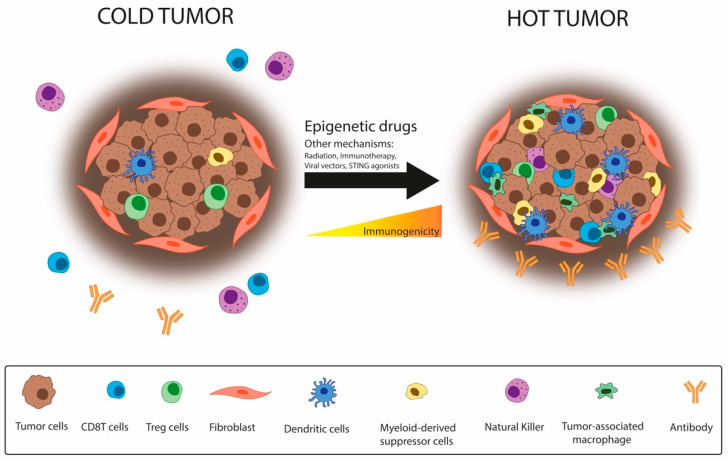
Epigenetic drugs contribute to turning cold tumors into hot. Cancer cells modulate their microenvironment to promote immune escape, increasing the presence of immunosuppressive cells on the tumor site and becoming a “cold tumor”, which displays a worse response against immunotherapy. Epigenetic drugs may switch this state, enhancing the immune response through the recruitment of CTLs, becoming a “hot tumor”. Hot tumors respond better to immunotherapy.

**Table 1 cancers-13-04139-t001:** Ongoing clinical trials of epigenetic drugs in TNBC.

Drugs	Identifier	Inclusion Criteria	Status	Patients (*n*)
Epigenetics drugs in combination with ICIs				
Entinostat (HDACi) Atezolizumab (anti-PD-L1)	NCT02708680	Adv. TNBC	Completed	88
Atezolzumab (anti-PD-L1) RO6870810 (BETi)	NCT03292172	TNBC	Terminated (Portfolio prioritization)	36
PRD001 (anti-PD-1) Panobinostat (HDACi) Other drugs	NCT02890069	TNBC + Other cancers	Recruiting	315
Epigenetics drugs alone or in combination with other treatments				
Entinostat (HDACi) +Other drugs	NCT04296942	Adv. TNBC or HER2+ BC	Recruiting	65
Decitabine (DNMTi) +Other drugs	NCT02957968	Locally Adv. HER2- BC	Recruiting	32
Romidepsin (HDACi) + Other drugs	NCT02393794	Locally recurrent/metastatic TNBC and/or with BRCA1/2 mutation	Suspended	54
Entinostat (HDACi)	NCT03361800	Early-stage TNBC	Terminated (Funding withdrawn)	5
Entinostat (HDACi) + Anastrozole	NCT01234532	Early-stage TNBC	Terminated (Low accrual)	5
Entinostat (HDACi) + Azacitidine (DNMTi)	NCT01349959	Adv. HER2-BC	Active, not recruiting	58
Decitabine (DNMTi) + Other drugs	NCT01194908	Adv.TNBC	Terminated (Slow accrual)	5
Decitabine (DNMTi) + Carboplatin	NCT03295552	Metastatic TNBC	Recruiting	59
Belinostat (HDACi) + Ribociclib	NCT04315233	Metastatic TNBC Recurrent ovarian cancer	Recruiting	34
Panobinostat (HDACi) + Letrozole	NCT01105312	Metastatic TNBC	Completed	28
Chidamide (HDACi)	NCT04582955	Early-stage TNBC	Recruiting	20
Chidamide (HDACi) + Cisplatin	NCT04192903	Relapsed or Metastatic TNBC	Not yet recruiting	55
Birabresib (BETi)	NCT02698176	Adv.TNBC + Other cancers	Terminated (Limited efficacy)	13
Birabresib (BETi)	NCT02259114	Adv.TNBC + Other cancers	Completed	47

Abbreviations: Adv: Advanced. HDACi: HDAC inhibitor. DNMTi: DNMT inhibitor. BETi: BET inhibitor. BC: Breast Cancer.

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
