# Peer review of "Epigenetic Regulation of Immunotherapy Response in Triple-Negative Breast Cancer"

_cancers, 2021, doi:10.3390/cancers13164139_

Round 1

Reviewer 1 Report

Pere Llinàs-Arias and colleagues provide a very clear and comprehensive review article on epigenetic changes in triple negative breast cancer and their contribution to the cancer cell–immunity cycle.

The article is well-written and comprehensible. The figures are really well presented, compliments!

I have only a minor suggestion:

The authors should add a paragraph regarding subtypes withing triple negative breast cancer. In specific, basal-like 1 (BL1), basal-like 2 (BL2), mesenchymal (M), mesenchymal stem–like (MSL), immunomodulatory (IM), and luminal androgen receptor (LAR).

Author Response

Thank you for your comments. We are glad to hear that you found our review comprehensive. In our introduction section, we have added a paragraph talking about the TNBC subtypes described by Lehmann et al. (Page 2). 

Reviewer 2 Report

Dear Author,

The review article  focused on Epigenetic regulation of immunotherapy response in TNBC. This article mainly focused how epigenetics strengthen immunotherapy by checkpoint inhibitor. As we know TNBC is aggressive and poor prognosis compared to other types of breast cancer. This article highlighted the mechanism of immunotherapy-based treatment in TNBC and summarizing the current clinical trials using epigenetic drugs and combined with immune-checkpoint inhibitor or small molecules.

I have few suggestions regarding the article. 

The minor changes are                 

1. The author highlighted cancer-immunity cycle proposed by Chen and Mellman. However, this review article focused few sections on epigenetic regulation of cancer-immunity cycle in TNBC. It will better author can provide more detail about antigen presentation, T cell priming and activation and T cell trafficking to blood vessels.

2. I have a suggestion to the author change the paragraph number from 2.5 to 3. So, in that section author can give sub-heading 3.1 Tumor microenvironment, 3.2 Metabolic rewiring and 3.3 will be EMT. Following that change the heading numbers accordingly.

Thanks.

Author Response

Reviewer #2 (COMMENTS FOR THE AUTHORS):

Dear Author,

The review article focused on Epigenetic regulation of immunotherapy response in TNBC. This article mainly focused how epigenetics strengthen immunotherapy by checkpoint inhibitor. As we know TNBC is aggressive and poor prognosis compared to other types of breast cancer. This article highlighted the mechanism of immunotherapy-based treatment in TNBC and summarizing the current clinical trials using epigenetic drugs and combined with immune-checkpoint inhibitor or small molecules.

I have few suggestions regarding the article. 

The minor changes are                 

  1. The author highlighted cancer-immunity cycle proposed by Chen and Mellman. However, this review article focused few sections on epigenetic regulation of cancer-immunity cycle in TNBC. It will better author can provide more detail about antigen presentation, T cell priming and activation and T cell trafficking to blood vessels.

Response: Thanks for reviewing our manuscript and making suggestions that improve the quality of the review.

We introduced the cancer-immunity cycle as a guide for the reader to follow the content of this section. As the reviewer indicates, the relevance of epigenetics is discussed only in some steps of the cycle described by Chen and Mellman. Specifically, we have focused on those steps where the epigenetic changes involved in immune response occur in TNBC cells. Therefore, as suggested by the reviewer, we have expanded the section describing the epigenetic mechanisms involved in antigen presentation on tumor cells and provided references to guide the reader on the mechanisms involved in dendritic cells (DC) and T cell maturation and activation (Pages 4 and 5).

  1. I have a suggestion to the author change the paragraph number from 2.5 to 3. So, in that section author can give sub-heading 3.1 Tumor microenvironment, 3.2 Metabolic rewiring and 3.3 will be EMT. Following that change the heading numbers accordingly.

Response: Thanks for this suggestion. We have modified the manuscript sections accordingly.